# IDH Mutations in Glioma: Molecular, Cellular, Diagnostic, and Clinical Implications

**DOI:** 10.3390/biology13110885

**Published:** 2024-10-30

**Authors:** Kristian A. Choate, Evan P. S. Pratt, Matthew J. Jennings, Robert J. Winn, Paul B. Mann

**Affiliations:** 1Upper Michigan Brain Tumor Center, Northern Michigan University, Marquette, MI 49855, USA; kchoate@uw.edu (K.A.C.); evpratt@nmu.edu (E.P.S.P.); majennin@nmu.edu (M.J.J.); rwinn@nmu.edu (R.J.W.); 2Department of Chemistry, Northern Michigan University, Marquette, MI 49855, USA; 3School of Clinical Sciences, Northern Michigan University, Marquette, MI 49855, USA; 4Department of Biology, Northern Michigan University, Marquette, MI 49855, USA

**Keywords:** isocitrate dehydrogenase, *IDH1*, *IDH2*, D-2-Hydroxyglutarate, D-2-HG, epigenetics, glioma

## Abstract

In this review, we discuss the role of isocitrate dehydrogenase in a normal biological context and its subsequent role in oncogenesis when certain mutations are acquired. In particular, we focus on the many downstream effects of the oncometabolite D-2-Hydroxyglutarate, the byproduct of mutant isocitrate dehydrogenase. D-2-Hydroxyglutarate interferes with essential biological pathways that result in significant alterations to epigenetics, metabolism, RNA transcript stability, and DNA damage repair. Additionally, we review the diagnostic methods available for detecting isocitrate dehydrogenase mutations and/or D-2-Hydroxyglutarate. The clinical implications, including the classification of isocitrate dehydrogenase mutants in glioma and pharmaceutical inhibitors, are also discussed.

## 1. Introduction

Gliomas are brain tumors associated with high mortality and life-altering symptoms including seizures, cognitive/motor deficits, dysphagia, and aphasia [1]. Current treatment methods for glioma include surgical resection, radiation therapy, and chemotherapy with the use of temozolomide (TMZ). Despite these approaches, the long-term survival of patients remains poor. Approximately 30% of primary brain tumors are gliomas, which are believed to arise from neuroglial stem or progenitor cells [2]. Gliomas may be categorized as astrocytomas, oligodendrogliomas, oligo-astrocytoma, or ependymomas depending upon the cell type(s) from which they originate (Figure 1). When classifying these tumors, the World Health Organization Central Nervous System 5 (WHO CNS5) grades ranging from I to IV are traditionally used, with grade I tumors being the least malignant [3]. To determine tumor grade, characteristics such as invasiveness, rate of growth, and degree of necrosis within the tumor are utilized. Gliomas may be diffuse in nature, which causes difficulty in both visualization via magnetic resonance imaging (MRI) and determination of margins during surgery, further complicating the identification and removal of malignant tissue.

Molecular analysis of the tumor allows for further classification of gliomas into subgroups. The classification of tumors provides valuable insight toward prognosis, and in some cases, the possibility of treatment with targeted therapies. Molecular classification may be accomplished with techniques including DNA/RNA sequencing, PCR, and DNA methylome profiling [4,5,6]. Various biomarkers have been established through the molecular classification of tumors, an example of which is isocitrate dehydrogenase (*IDH*). *IDH* enzymes play essential roles in metabolic processes such as the citric acid cycle, lipogenesis, glutamine metabolism, and redox regulation [7]. Oncogenic mutations to *IDH1* were identified in 2008 following an integrated genomic analysis of glioblastoma multiforme (GBM) [8], and have since been targeted with pharmaceutical inhibitors [9,10]. The prognosis and treatment for *IDH* mutant gliomas differ from their *IDH* wildtype counterparts significantly, such that screening for them has become an important standard of care. Mutations to *IDH* have been documented in various types of cancer [11,12,13] but are more prevalent in glioma and acute myeloid leukemia (AML) [14,15]. In general, *IDH* mutations are associated with less aggressive cancer, as they render cells more vulnerable to death [16] and demonstrate conservative levels of migration, angiogenesis, and invasion [17]. In this review, we describe the molecular, cellular, and clinical consequences of *IDH* mutations in glioma as well as the current diagnostic methods and treatments.

## 2. Normal Function of IDH and the Cancer-Associated Accumulation of D-2-HG

*IDH1/2* are the most frequently mutated cancer-associated metabolic genes. IDH exists in three isoforms: IDH1, IDH2, and IDH3. IDH1 and IDH2 enzymes function as homodimers and share approximately 70% sequence homology [18]. Conversely, IDH3 is a more distantly related heterotetrameric protein that is not known to be associated with the development of cancer [19]. *IDH1* is expressed in the cytosol and peroxisomes and produces NADPH for fatty acid biosynthesis, with oncogenic mutations commonly associated with glioma. Alternatively, IDH2 localizes within the mitochondrial matrix, with mutations often driving AML, or less commonly, glioma [20]. *IDH1* and *IDH2* mutations are mutually exclusive, with rare exceptions [21], due to a common biochemical mechanism [22] that drives similar downstream effects. Ordinarily, IDH1/2 catalyze the conversion of isocitrate into *α*-ketoglutarate (*α*KG) while reducing NADP^+^ to NADPH as a byproduct. Missense mutations yield amino acid changes at positions R132 in *IDH1* and R140 or R172 in *IDH2*. These mutations result in the replacement of conserved arginine residues in the catalytic site with an alternative amino acid. The consequence of these mutations is an altered protein with neomorphic activity—the conversion of *α*KG to D-2-Hydroxyglutarate (D-2-HG) [14,23] (Figure 2). D-2-HG is an oncometabolite that accumulates in primary *IDH* mutant tumors at concentrations ranging from 1 to 30 mM [14,23]. D-2-HG causes a plethora of downstream effects that mediate the development of cancer, as detailed below.

## 3. Epigenetic Alterations

Epigenetic alterations are a hallmark of *IDH* mutant-mediated oncogenesis and are largely driven by the production of D-2-HG. Due to its structural similarity to *α*KG and accumulation within *IDH* mutant cells, D-2-HG competitively inhibits various *α*KG-dependent dioxygenases. These dioxygenases include the Jumonji-C (JmjC) family of histone lysine demethylases (KDMs), ten-eleven translocation (TET) DNA cytosine-oxidizing enzymes, AlkB homologs (ABH), and prolyl hydroxylases (PHD) [24] (Figure 3). *α*KG-dependent dioxygenases play critical roles in epigenetics [24], biosynthesis [25], post-translational modifications [26], and hypoxic response [27]. The collective antagonization of demethylases by D-2-HG leads to instability of RNA transcripts and characteristic patterns of hypermethylation of DNA and histone proteins [28]. Due to these epigenetic modifications, mutant *IDH* is highly associated with the Glioma CpG island methylator phenotype (G-CIMP) [29], a locus-specific pattern of DNA hypermethylation at CpG-rich promoter regions. 

### 3.1. JmjC-KDMs

JmjC-KDMs are enzymes that regulate gene expression through demethylation and chromatin scaffolding functions [30]. The JmjC protein family contains >30 members [31], several of which may actively contribute to the development of cancers including glioma [32], acute myeloid leukemia [33], and lung cancer [31] when dysregulated. Numerous silenced, mutated, or deleted JmjC-KDM genes have been identified in cancer [34] and appear to act as either a driver or suppressor of gene expression, depending upon the cellular context. Since JmjC-KDMs aid in the control of genome-wide expression patterns, aberrant protein function may result in significant changes to the transcriptomic profile of the cell [35]. Potential targets of dysregulated JmjC-KDMs include pro-inflammatory pathways such as signal transducer and activator of transcription (STAT)-dependent response or proliferation-associated pathways such as mitogen-activated protein kinase (MAPK) [31]. 

In some cases, the D-2-HG-mediated inhibition of JmjC-KDMs results in downstream effects that function in a manner similar to knockout models. This phenomenon has been demonstrated by several members of the JmjC-KDM5 family of H3K4 histone lysine demethylases (KDM5A, KDM5C, and KDM5D). The inhibition of these JmjC-KDM5 enzymes occurs at physiologically relevant concentrations of D-2-HG (less than 1 mM) and is thought to contribute to *IDH* mutant-mediated transformation [36]. Of all potential *α*KG analogs (for example, fumarate, L-2-HG, succinate), D-2-HG is the most potent inhibitor of KDM4A/B and KDM6A with IC_50_ values < 200 µM. Collectively, KDMs inhibited by D-2-HG contribute to the regulation of diverse cellular processes including differentiation [37], growth and development [37,38], adipogenesis [39], adhesion, and proliferation [33]. The competitive inhibition of several JmjC-KDMs by D-2-HG has been shown to increase dimethylation of H3K79, H3K27, and H3K9 as well as mono and trimethylation of H3K4 [40]. In combination with high levels of acetylation, H3K79 and H3K4 methylation are associated with active chromatin, with H3K79 found within the coding regions of genes and H3K4 often enriched at enhancers and promoters of actively transcribed genes. Conversely, H3K27 and H3K9 are associated with inactive chromatin [41]. Thus, the pleiotropic effects of D-2-HG accumulation impact a variety of different biochemical pathways resulting in oncogenesis.

Although the epigenetic changes mediated by JmjC-KDMs in *IDH* mutant cancers are profound, they are also readily reversible by the addition of *α*KG. A total of 300 µM *α*KG was shown to negate the inhibition of KDM7A caused by up to 50 mM D-2-HG [40]. This reversibility demonstrates the relatively weak antagonism of JmjC-KDMs by D-2-HG despite occupying the same active site in the catalytic core. Regardless, *IDH1*-R132H mutations have shown a near 60% reduction of *α*KG and >20-fold increase in D-2-HG [40], suggesting that this competitive inhibition is due to metabolite quantity rather than changes to enzyme kinetics. Despite the inhibition of several JmjC-KDMs by D-2-HG, it is important to note that *α*KG-dependent dioxygenases vary greatly in their susceptibility to inhibition by D-2-HG, with some not significantly affected by relevant concentrations [34], such as KDM5B (IC_50_ 3.6 mM) [42]. Recently, pharmaceutical inhibitors have been screened for their therapeutic potential toward inhibiting relevant JmjC-KDMs *IDH* mutant cancers, which we address later in this review.

### 3.2. Histone Deacetylases

Histone deacetylases (HDACs) possess chromatin remodeling activity and contribute to silencing genes in an epigenetic manner. HDACs hydrolyze acetate from lysine on histone tails [43], strengthening the interaction between histone proteins and DNA. HDACs are recruited to methylated DNA through their association with methyl-CpG binding proteins [44]. As hypermethylation is abundant in IDH mutant cancers, it is thought that HDACs play a role in transcriptomic alterations by compounding the effects of methylation to further condense chromatin near CpG islands. Expression profiles of *IDH* mutant glioma have demonstrated the upregulation of genes promoting HDAC function or related pathways in comparison to their *IDH* wildtype counterparts, including three of the six HDAC genes that are expressed in gliomas [45]. Additional studies have shown that overexpression of the IDH mutant enzyme results in histone hypoacetylation [46], further supporting the notion that HDACs play a role in *IDH* mutant oncogenesis. Several groups have presented HDAC inhibitors as a potential therapeutic for *IDH* mutant glioma, which we will discuss later in this review.

### 3.3. TET Enzymes

TET enzymes are involved in various biological processes including post-transcriptional regulation [47] and are essential for immune cell development [48] and embryogenesis [49]. TET enzymes generate oxidized derivatives of 5-methylcytosine (5 mC) in a Fe(II)/*α*KG-dependent manner. In these reactions, 5mC is sequentially oxidized to 5-hydroxymethylcytosine, 5-formylcytosine, then 5-carboxylcytosine. This oxidation typically occurs at CpG dinucleotide sites, where thymine DNA glycosylase excises 5-formylcytosine and 5-carboxylcytosine then base excision repair is utilized to yield unmethylated cytosine [47,50]. Rather than being inert intermediates, the oxidation products of 5 mC play unique and distinct biological roles and may also actively contribute to cancer in some cases [51]. For example, Johnson et al. observed the enrichment of 5-hydroxymethylcytosine in GBM at disease-specific enhancer elements to unexpectedly elicit open chromatin and increased gene expression [52]. To our knowledge, no data are currently available that explore the relationship between IDH mutant cancers and 5 mC oxidation products. 

The dysregulation of TET enzymes contributes to the development of the G-CIMP through an increase in methylation [53] and subsequently alters the transcriptomic profile of *IDH* mutant cancers. Similar to JmjC-KDMs, TET enzymes are often mutated in various types of cancer such as AML and myelodysplastic/myeloproliferative neoplasms [54,55]. TET and IDH driver mutations are mutually exclusive [56], suggesting that the inhibition of TET enzymes contributes significantly to mutant *IDH*-mediated oncogenesis [46]. While TET or *IDH* mutations routinely occur in leukemia [47,57], TET-mediated oncogenesis is not typically thought to contribute to the development of glioma. TET enzymes are susceptible to competitive inhibition by the accumulation of D-2-HG, fumarate or succinate that result from oncogenic mutations to *IDH*, fumarase hydratase or succinate dehydrogenase, respectively [58]; however, L-2-HG, a naturally occurring enantiomer of D-2-HG that is upregulated under hypoxic conditions [59], does not appear to interfere with TET activity intracellularly. To this end, the use of cell-permeable L-2-HG to quantify TET enzymatic inhibition demonstrates that treatment with up to 3 mM of L-2-HG elicits little to no inhibition of TET2 or TET3 in situ [60]. 

Despite the oncogenic consequences of TET dysregulation, not all members of the TET family are prone to inhibition by clinically relevant concentrations of D-2-HG. For example, TET2 has demonstrated inhibition at low levels of D-2-HG in vitro (IC_50_ = ~15 µM) while TET1 (IC_50_ = ~800 µM) and TET3 (IC_50_ = ~100 µM) are less susceptible. Although TET2 is the most prone to competitive inhibition by D-2-HG, TET1 and TET3 are highly expressed in specific areas of the brain including the cerebellum and cerebral cortex, respectively, while the expression of TET2 in the brain is somewhat modest [47]. Another potential variable regarding the inhibition of TET enzymes in *IDH* mutant cancers is the availability of ascorbic acid (vitamin C). Ascorbic acid modulates TET enzymes via a direct interaction with their catalytic domain [61] and reduces Fe(III) to Fe(II), an essential cofactor of TET. Indeed, co-administration of pharmaceutical mutant *IDH* inhibitors and ascorbic acid has been shown to restore TET activity in *IDH1*-R132H colorectal cancer cell lines [62]. Additional studies have demonstrated that supplementation with an oxidatively stable form of vitamin C (Ascorbate-2-phosphate) can restore 2-HG-induced epigenetic remodeling [63]. Because *IDH* mutations alter NADP^+^/NADPH ratios and NADPH is a cofactor in recycling ascorbate, it is possible that vitamin C availability and D-2-HG accumulation have a combinatorial effect towards the dampened TET activity witnessed in *IDH* mutant cancers. To this end, an in vivo knock in model of *IDH1*-R132H has demonstrated altered NADP^+^/NADPH ratios with a coinciding decrease in ascorbate within the brain [64]. 

## 4. Prolyl Hydroxylases and Hypoxic Response

In addition to epigenetic modifications, *α*KG-dependent dioxygenases may also play key roles in facilitating decisions regarding cell fate such as normal differentiation [65] and regulation of hypoxia-inducible factor 1 (HIF-1*α*) [66]. Due to the rapid proliferation of tumors, hypoxia is frequently observed in the microenvironment as existing blood supplies are outgrown. HIF-1*α* is a transcription factor that is upregulated in response to low oxygen, including under inflammatory conditions [67], to induce the expression of various genes essential for cell survival and adaptation to a hypoxic environment [68]. PHDs utilize *α*KG as a substrate to selectively hydroxylate targets and are susceptible to dysregulation by the accumulation of D-2-HG [69]. Under ordinary conditions, HIF-1*α* is constitutively expressed and rapidly degraded by PHDs. Low levels of oxygen inhibit PHDs, resulting in elevated expression of HIF-1*α* and subsequently the upregulation of hypoxia-associated genes. 

EglN prolyl-4-hydroxylases ordinarily mark HIF-1*α* for polyubiquitylation and proteasomal degradation [70]; however, literature disagrees on whether D-2-HG stimulates or inhibits these PHDs. Some studies suggest that D-2-HG acts as a co-substrate to PHDs resulting in the destabilization of HIF-1*α* and reduced expression of associated genes [71]. The knockdown of PHDs in *IDH* mutant astrocytes was also shown to result in a marked reduction in proliferation, suggesting a potential role for HIF-1*α* as a tumor suppressor [69]. As ascorbic acid availability is lowered by the altered NADP+/NADPH ratios in *IDH* mutant cancers [64], it should also be noted that ascorbate encourages hydroxylase activity to suppress the transcriptional response of HIF-1*α* [72,73]. To this end, a decrease in ascorbate availability could also contribute to a decrease in the expression of hypoxia-associated pathways; however, this relationship has never been explored. 

Contrary to findings that HIF-1*α* is downregulated in *IDH* mutant cancers, many others have shown that HIF-1*α* is upregulated in *IDH* mutant cells, cells treated with exogenous 2-HG, *IDH* mutant tumors, and brains of mouse embryos expressing *IDH1*-R132H [40,64,74,75]. The notion of an increase in HIF-1*α* resulting from metabolically driven oncogenesis such as *IDH* is supported by cancers driven by a similar mechanism. Oncogenic mutations to fumarase hydratase or succinate dehydrogenase result in the accumulation of fumarate or succinate, respectively, which act in a similar manner to D-2-HG by competitively inhibiting *α*KG-dependent dioxygenases. In these instances, fumarate or succinate have been shown to inhibit *α*KG-dependent PHDs that target HIF-1*α* for degradation [76,77], resulting in its subsequent accumulation. 

## 5. RNA Transcript Stability

The methylation of adenosine at the nitrogen-6 positions of RNA (N^6^-Methyladenosine, m^6^A) serves as an essential posttranslational regulatory mark of mRNAs and noncoding RNAs. In addition to playing a role in most RNA-related processes such as splicing and nuclear export, m^6^A is also associated with biological processes including transcriptional regulation, signal transduction, DNA damage response [78], and cancer [79]. m^6^A is a frequent internal modification of mRNA which recruits m^6^A binding proteins such as YTHDF1/2/3, YTHDC2, or IGF2BP1/2/3 to regulate stability and/or translation efficiency [79]. m^6^A modifications are ordinarily facilitated by methyltransferases and removed by demethylases such as fat mass and obesity-associated protein (FTO) or ALKBH5 which are Fe(II)/*α*KG-dependent dioxygenases [80]. 

*IDH* wildtype cells treated with D-2-HG or *IDH* mutant cells producing D-2-HG exhibit an increase in m^6^A levels that can be reversed through treatment with *IDH* mutant pharmaceutical inhibitors such as Vorasidenib [81]. Since IDH mutant pharmaceutical inhibitors are known to deplete D-2-HG, these results support the role of the oncometabolite toward m^6^A accumulation. Moreover, D-2-HG has demonstrated a dose-dependent inhibition of FTO [81] that causes an increase in m^6^A levels [82,83,84]. The knockdown of endogenous FTO recapitulates the impacts of D-2-HG on cell viability, increases m^6^A levels [83], and promotes apoptosis and cell-cycle arrest at the G_0_/G1 phase [84]. Similarly, pharmacologic inhibition of FTO using FB23-2 or meclofenamic acid results in decreased proliferation and increased apoptosis. In addition to FTO, D-2-HG also inhibits ALKBH5; however, D-2-HG has a weaker binding affinity for ALKBH5 than FTO and the knockdown of ALKBH5 alone does not recapitulate the increase in m^6^A seen in FTO knockout lines or *IDH* mutant cells [81]. These results suggest that FTO inhibition is the primary mechanism of m^6^A accumulation in *IDH* mutant cells.

Increased levels of m^6^A result in lower stability of mRNA transcripts with a notable target of D-2-HG mediated transcript decay being the oncogene *MYC* [83]. Interestingly, ectopically expressing MYC rescues D-2-HG-mediated growth inhibition [84]. Further studies have shown that m^6^A-mediated downregulation of activating transcription factor 5 (*ATF5*) mRNA may also play an important role in regulating proliferation and apoptosis in *IDH* mutant glioma [81]. In addition to impacting proliferation, m^6^A levels modify the metabolic profile of cells, with the largest impacts being the downregulation of phosphofructokinase platelet (PFKP) and lactate dehydrogenase B (LDHB) [84].

While TET enzymes are largely known for facilitating the active demethylation of DNA, recent studies have also demonstrated that TET2 contains an RNA-binding domain [85] and can oxidize 5-methylcytosine RNA into 5-hydroxymethylcytosine. Since 5-hydroxymethylcytosine is associated with active translation of mRNA [86], aberrant post-translational mRNA modifications by TET2 reduce transcript stability which may increase susceptibility to certain diseases [87]. With these recent findings, further studies are necessary to determine the specific impact of TET-mediated mRNA alterations in *IDH* mutant glioma. 

## 6. Patterns of Transcription

The many epigenetic alterations within the *IDH* mutant genome have various downstream implications for the transcriptome. Tran et al. assessed RNA sequencing (RNAseq) and microarray data for 1032 gliomas from the cancer genome atlas and 395 gliomas from REMBRANDT and found four distinct transcriptomic profile groups. Interestingly, *IDH* mutant gliomas with codeletions were grouped with oligodendrogliomas with high tumor purity. Transcriptomic data for this group was enriched for neurotransmission, G-protein coupled receptor signaling, and insulin secretion pathways. Alternatively, *IDH* mutant gliomas without codeletions generally corresponded with astrocytoma and transcriptomic data enriched for immune, cell cycle, NOTCH signaling, transcription, and translation [88]. In a similar study, Cheng et al. utilized RNAseq data to establish a six-gene risk signature for *IDH* mutant low-grade glioma to assist in determining risk and prognosis. The six genes found to be highly significant for prognosis in *IDH* mutant patients included: cell division cycle (CDC) 20, Wiskott–Aldrich syndrome protein family (WASF)3, deleted in breast cancer (DBC)1, engrailed (EN)2, vimentin (VIM), and carboxypeptidase (CPE). Higher expression of CDC20, EN2, and VIM was found to be associated with risk while WASF3, DBC1, and CPE were considered protective genes with higher expression levels associated with better prognosis [89]. Another study focusing on transcriptomic profiles compared *IDH* mutant gliomas and *IDH* mutant AML, melanoma, and cholangiocarcinoma. Interestingly, these profiles showed pro-malignant genes unique to *IDH* mutant gliomas while genes associated with differentiation and immune response were suppressed in all *IDH* mutant cancers. Moreover, *IDH* mutant gliomas demonstrated a higher degree of genes that were both hypermethylated and differentially expressed in comparison to other types of *IDH* mutant cancers. Unruh et al. also noted variances in differential gene expression between *IDH* mutant oligodendrogliomas and astrocytoma, with oligodendrogliomas downregulating genes linked to angiogenesis, cell proliferation, and integrin binding [90]. Additionally, analysis of transcriptomic profiles of *IDH* mutant glioma patients revealed the decreased expression of HIF-1*α* targets as well as the inhibition of angiogenesis and vasculogenesis. Specifically, the expression of EGLN1 and EGLN3 PHDs, which serve to degrade HIF-1*α* was upregulated while pro-angiogenic targets such as vascular endothelial growth factor A, angiopoietin-2 and platelet-derived growth factor A were decreased [27].

## 7. Metabolic Reprogramming in *IDH* Mutant Glioma

### 7.1. Comparison of Wildtype and Mutant IDH Reactions

Wildtype IDH catalyzes the oxidative decarboxylation of isocitrate into αKG and CO_2_. This reaction occurs concomitantly with the reduction of NADP^+^ into NADPH, with the exception of IDH3, which generates NADH within the citric acid cycle [91]. Under certain circumstances (e.g., hypoxia), IDH can catalyze the reverse reaction—the reductive carboxylation of αKG into isocitrate, generating NADP^+^ as a byproduct [92]. Isocitrate is subsequently isomerized into citrate, which is broken down by ATP citrate lyase into acetyl-CoA. Acetyl-CoA can be used for fatty acid biosynthesis under these conditions [93]. 

Hotspot mutations within the active site of IDH1 (R132X) and IDH2 (R140X or R172X) render the enzymes capable of catalyzing a new reaction—the conversion of αKG into D-2-HG. This is in contrast to the reverse reaction catalyzed by wildtype IDH1 that produces isocitrate. Mutant *IDH*-driven D-2-HG accumulation is coupled with the oxidation of NADPH into NADP^+^. Unlike wildtype IDH, mutant IDH consumes both αKG and NADPH, which is likely to impact cellular metabolism and redox homeostasis, respectively. 

D-2-HG is classified as an oncometabolite and is almost exclusively produced by *IDH* mutant cells [14], although there is evidence that other enzymes, including phosphoglycerate dehydrogenase, produce minute amounts of the oncometabolite in cells [94]. Mutant IDH-mediated D-2-HG accumulation is a molecular hallmark of astrocytoma and oligodendroglioma [95]. Importantly, D-2-HG production drives metabolic rewiring in these types of cancer, including but not limited to dysregulation of glucose metabolism, the citric acid cycle, amino acid metabolism, lipid and cholesterol metabolism and redox homeostasis. 

### 7.2. Mutant IDH Cells Do Not Perform Aerobic Glycolysis

Perhaps the most well-known example of metabolic reprogramming is the overconsumption of glucose by cancer cells to meet growing energy demands [96]. Cancer cells are also known to produce and secrete copious amounts of lactate as a result of pyruvate metabolism, despite the presence of O_2_. This phenomenon is better known as aerobic glycolysis or the Warburg effect. Interestingly, *IDH* mutant glioma does not appear to follow this framework. Glycolytic flux is reduced in *IDH1* mutant glioma tumors compared with *IDH1* wildtype tumors as a result of dampened expression of the rate-limiting glycolytic enzymes hexokinase and pyruvate kinase [97]. In agreement with this, lower quantities of glycolytic intermediates, including fructose 1,6-bisphosphate, 3-phosphoglycerate and phosphoenolpyruvate, have been observed in *IDH1* mutant glioma tissue [95].

*IDH* mutant glioma tissue and brain tumor stem cells derived from *IDH1* mutant glioma tissue show reduced expression of lactate dehydrogenase A (LDHA), the enzyme responsible for converting pyruvate into lactate [98]. Silenced LDHA expression is associated with hypermethylation of the *LDHA* promoter, which is likely to be a direct result of inhibition of αKG-dependent DNA demethylases by D-2-HG. This relationship, along with other IDH-mutant-specific metabolic and epigenetic aberrations, is summarized in Figure 4. This is accompanied by decreased expression of monocarboxylate transporters (MCT) 1 and 4, both of which are involved in the efflux of lactate from the cell. Overall, this suggests that *IDH1* mutant cells do not perform aerobic glycolysis. Indeed, lactate levels are undetectable in *IDH1* mutant neurospheres [99]. Interestingly, the enzyme LDHB, which is involved in converting lactate to pyruvate, and the lactate importer MCT2 are more abundant in *IDH1* mutant glioma tissue compared with wildtype samples [100]. This suggests that lactate may serve as an anaplerotic substrate for pyruvate accumulation.

### 7.3. Citric Acid Cycle Rewiring in IDH Mutant Cells

The citric acid cycle is the cornerstone of cellular metabolism [101]. This pathway is directly involved in energy production and supplies metabolic intermediates for the generation of fatty acids, amino acids and nucleotides. Although IDH3 is the only IDH isozyme that directly participates in the citric acid cycle, both IDH1 and IDH2 help regulate levels of cellular αKG and control redox homeostasis. There is strong agreement among various studies that the citric acid cycle is rewired in *IDH1* mutant gliomas. Elevated expression of citric acid cycle enzymes upstream of IDH, including citrate synthase and aconitase, has been reported [97]. In addition, there is decreased expression of enzymes downstream of IDH, including succinate dehydrogenase and fumarase. These findings are supported by studies showing that citric acid cycle intermediates downstream of IDH, including αKG, succinate and fumarate, are decreased [102,103].

Anaplerotic reactions serve to replenish citric acid cycle intermediates, maintaining cycle flux [101]. Three common anaplerotic nutrients are glutamate (converted to αKG by glutamate dehydrogenase), glutamine (converted to glutamate by glutaminase) and pyruvate (converted to oxaloacetate by pyruvate carboxylase). As αKG is consumed by mutant IDH, the metabolism of both glutamate and glutamine would be expected to be particularly important. Indeed, both glutamate and glutamine levels are significantly decreased in U87MG cells expressing the IDH1-R132H mutant, compared with wildtype cells [104]. In partial agreement with this finding, patient-derived *IDH1*-R132H xenografts showed elevated expression of glutamate dehydrogenase and lower levels of glutamate in mouse brains, suggesting that glutamate metabolism is increased in *IDH1* mutant tissue [97,105,106]. However, the authors found no significant difference in the expression of glutaminase or cellular glutamine levels. Somewhat surprisingly, another study found no differences in glutamate or glutamine levels between *IDH1* wildtype and *IDH1* mutant human glioma tumor samples [97]. Thus, it appears that glutamate versus glutamine usage by *IDH* mutant cells is context-dependent and requires further exploration.

It is worth noting that pyruvate carboxylase expression was elevated in immortalized human astrocytes expressing the *IDH1*-R132H mutant and in human glioma tissue [104]. This suggests that regardless of whether αKG is primarily derived from glutamate or glutamine, pyruvate carboxylase may take the driver’s seat as the primary anaplerotic reaction in the citric acid cycle.

### 7.4. Mutant IDH Drives Changes in Amino Acid Metabolism

McBrayer et al. discovered that expression of *IDH1*-R132H in glioma cell lines elicits an upregulation in glutaminase expression [107]. Interestingly, the authors concluded that enhanced glutamine catabolism was being used to primarily support glutamate production, as opposed to αKG. Glutamate levels were decreased in the mutant cell line as a direct result of branched-chain aminotransferase 1 and 2 (BCAT1/2) inhibition by D-2-HG. As expected, the authors found that the mutant *IDH1* cells had increased levels of the branched-chain amino acids (BCAA) valine, isoleucine and leucine. Two additional studies support the finding that BCAT1/2 expression and activity are decreased in *IDH* mutant tumors compared with wildtype tumors [97,108]. However, Tonjes et al. suggested that reduced activity of BCAT1/2 could be attributed to lower levels of αKG due to mutant IDH activity. 

In addition to lower levels of BCAAs, *IDH1* mutant glioma tissue contains decreased concentrations of the amino acids glycine and serine [95]. Glycine and serine are integral to one-carbon metabolism, suggesting potential impairments in this metabolic network. In one-carbon metabolism, the interconversion of glycine and serine by the enzyme serine hydroxymethyltransferase is coupled with the folate cycle, the methionine cycle and nucleotide biosynthesis [109]. However, an in-depth analysis of these pathways in *IDH* mutant glioma is lacking.

### 7.5. Consumption of NADPH by the Mutant IDH Enzyme

NADPH is produced in the canonical reaction catalyzed by wildtype IDH, whereas mutant IDH consumes NADPH in the reaction that produces D-2-HG. As expected, U87MG glioma cells expressing the *IDH1*-R132H mutant and patient-derived xenografts from *IDH1* mutant GBM both showed lower cellular NADPH levels and a decreased [NADPH]/[NADP^+^] ratio [102,103]. Fack et al. found that [NADH]/[NAD^+^] was not different between *IDH1* wildtype and mutant tumors; however, Biedermann et al. detected less NAD^+^ in the mutant U87MG cells, suggesting that NAD^+^ could be used to replenish cellular NADPH. As a potential explanation for how redox homeostasis can be restored in *IDH* mutant cells, Hollinshead et al. found that human anaplastic oligodendroglioma cells expressing the IDH1-R132H mutant had increased expression of the enzyme proline 5-carboxylase reductase 1 (PYCR1), which is involved in proline biosynthesis [110]. The authors found that not only did the *IDH1* mutant cell line accumulate proline as a result of increased PYCR1 activity but NADH consumption in this reaction was a means to uncouple the citric acid cycle from NADH usage in oxidative phosphorylation.

The oxidative pentose phosphate pathway is recognized as the major producer of cytosolic NADPH. NADPH is primarily used for fatty acid biosynthesis and to maintain the cellular pool of reduced glutathione for mitigating ROS. Unfortunately, the impact of reduced NADPH levels in *IDH* mutant glioma cells specifically is largely lacking. In line with studies performed in glioma cells, HCT116 colon cancer cells expressing the IDH1-R132H mutant consume significantly greater quantities of NADPH compared with wildtype cells [111]. This leads to an increase in the cellular demand for NADPH, amplifying flux through the NADPH-producing pentose phosphate pathway. Interestingly, both *IDH1*-R132H mutants and *IDH2*-R172K mutants are more susceptible to oxidative insults. Gelman et al. also found that the *IDH1* mutation created competition between the production of D-2-HG and the fatty acid palmitate in colon cancer cells. Fibrosarcoma cells expressing another IDH1 mutant (R132C) consume NADPH at the same rate as de novo lipogenesis [112]. These cells have increased reliance on exogenous lipids for growth.

### 7.6. Dysregulation of Membrane Lipid Biosynthesis

The biosynthesis of fatty acids, membrane lipids and cholesterol is perturbed in *IDH1* mutant glioma. Three *IDH1* mutant glioma models, including human glioma xenografts in mice, cultured glioma cell lines and human glioma biopsies, showed a distinct phospholipid profile characterized by low levels of phosphoethanolamine and heightened levels of phosphatidylcholine [113]. In contrast, another study showed that phosphatidylcholine levels were decreased in U87MG cells expressing the IDH1-R132H mutation [114].

## 8. DNA Damage Repair

Various studies have demonstrated that cells with an *IDH* mutation have heightened sensitivity to DNA-damaging agents. This increased sensitivity is partially mediated by the inhibition of ABH enzymes including ALKBH2 and ALKBH3. ALKBH2/3 for the removal of alkyl adducts from nucleobases via oxidative dealkylation [115], and the inhibition of these enzymes by D-2-HG reduces the repair kinetics of *IDH* mutant cells resulting in the accretion of DNA damage. Since ABH proteins are *α*KG-Fe(II)-dependent dioxygenases, the accumulation of D-2-HG impedes normal function, resulting in a 73–88% inhibition of normal DNA repair activities [116]. Ratios of D-2-HG:*α*KG in patients have been observed at approximately 373:1, while D-2-HG concentrations are dependent upon the source, but in tissues can accumulate up to almost 2 mM [117]. For experiments gauging D-2-HG inhibition of DNA repair activities, Chen et al. utilized a 373:1 ratio of D-2-HG:*α*KG and concentrations of D-2-HG varying from 0–37 mM. Interestingly, like other dioxygenases that rely upon *α*KG, this inhibition is reversible if the *IDH* mutation (and consequently the production of D-2-HG) is lost [118]. The presence of excess *α*KG can also assist in reversing this inhibition [116].

In addition to ALKBH2/3, *IDH* mutant cells also demonstrate an impaired ataxia-telangiectasia-mutated (ATM) signaling pathway, which is an essential mediator of cellular response to double-stranded breaks (DSB) [119]. Upon the occurrence of a DSB, ATM is recruited to the damage site and then initiates DNA damage repair (DDR) complexes by phosphorylating various targets including checkpoint kinases and p53 to begin the repair process [120]. Further investigation into the mechanism of the *IDH* mutant-specific downregulation of ATM has demonstrated that the reduced expression of ATM is due to D-2-HG mediated histone methylation of H3-K9 by KDM4 [121]. The use of pharmaceutical inhibitor AGI-5198 with *IDH* mutant patient-derived cultured glioma cells has demonstrated the downregulation of essential epigenetic reader Zinc Finger MYND-Type Containing 8 (ZMYND8) [122]. ZMYND8 recognizes modifications such as acetylation and methylation, facilitates DNA repair in the presence of DSB, and may act as either a repressor or enhancer of transcription [123]. KDM5A-dependent H3K4me3 demethylation near DSB is required for the colocalization of ZMYND8 and subunits of the nucleosome remodeling (NuRD) and histone deacetylation complex including HDAC1/2 and chromodomain helicase DNA binding protein 3-5 (CHD3/4/5) [124]. Lending to the known sensitivity of *IDH* mutant cells to poly(ADP) ribose (PARP) inhibitors [125], it is also of note that ZMYND8/NuRD facilitated repair is dependent upon poly(ADP) ribose. To this end, treatment with PARP inhibitors has been shown to abolish ZMYND8 recruitment in cultured cells [124], and the knockdown of ZMYND8 in *IDH* mutant patient-derived cultured glioma cells resulted in increased sensitivity to radiotherapy as well as significant phosphorylation of ATM and γH2AX activation in response to ionizing radiation [122].

Despite several known mechanisms for increased vulnerability to DNA damage, variable results exist concerning the response of *IDH* mutant cells to radiation-induced DNA damage. In some cases, *IDH* mutant glioma cells have demonstrated a decreased sensitivity to radiation-induced DNA damage in comparison to *IDH* wildtype cells [126]. A potential theory for this unexpected finding posited by the authors is that *IDH* mutant cells must develop buffering mechanisms against high levels of ROS to survive, thus making them better equipped to survive ROS generated from radiation damage. Despite these findings, other groups have shown the opposite: that *IDH* mutations increase sensitivity to radiation due to delayed DNA repair [127,128]. For example, when treated with TMZ, *IDH* mutant cells have an IC_50_ less than half that of *IDH* wildtype cells [119]. Since the mechanism of TMZ is to disrupt DNA structure via the addition of alkyl groups, this heightened sensitivity suggests a weakened DDR response. 

Conflicting results regarding the response of *IDH* mutant cells to DNA damage could potentially be explained by the genetic complexity found within the tumor environment that varies between patients and is often not accounted for with isogenic knockout models. To this end, several mutations that commonly co-occur with *IDH* mutations [specifically inactivating *TP53* [129] and alpha thalassemia/mental retardation syndrome X-linked gene (*ATRX*) [130] have been shown to contribute to genomic stability, enhanced DNA repair, and resistance to genotoxic therapies [131]. Thus, although this relationship has not been extensively explored, it is possible that discrepancies in DDR data are due to co-occurring deletions or other complex genetic factors.

## 9. Immunological Impact of *IDH* Mutations in Glioma

The immunological implications of *IDH* mutations in glioma have recently been extensively covered [132]. 

## 10. Diagnostics

Efforts to identify and classify biomarkers have yielded valuable targets for diagnostics treatment, with IDH mutant cancers being a key example in glioma. Techniques that may elucidate the IDH mutational status prior to surgical resection remain urgent, as overall survival improves with a maximal surgical resection [133]. Still, CNS malignancies pose unique challenges, with the acquisition of tissue remaining at the forefront. While tissue-based assays remain valuable, many cannot function in a manner rapid enough to provide information capable of guiding the extent of surgical resection. However, strategies leveraging D-2-HG as a surrogate biomarker show great promise for facilitating early detection and monitoring of disease state. Additionally, MRI-based techniques have been highly successful for the early detection of IDH mutant gliomas. Here, we discuss some of the molecular diagnostic techniques developed for the detection of IDH mutations.

### 10.1. Sequencing

The development of assays to successfully detect the *IDH* mutant genotype has been challenging, particularly due to the heterogeneity of gliomas [134,135,136] and heterozygosity of *IDH* mutations [137], which collectively lend to a low copy number within a sample. Currently, the sequencing of DNA serves as one of the most comprehensive and relied-upon methods for characterizing the molecular profile of tumors. Sequencing is particularly beneficial as it is adept at identifying a variety of genetic abnormalities including single nucleotide variants (SNV), indels, and structural variants. The detection of SNVs is particularly relevant for identifying *IDH* mutations, as traditional PCR or other amplification-based methods struggle to identify the single base changes needed to differentiate between the wildtype and mutant genotype. Furthermore, there is evidence that sequencing-based detection is a more reliable indication of *IDH* mutational status than IHC [138], which is currently considered a gold standard. 

Next-generation sequencing (NGS) and third-generation sequencing platforms are advantageous as they are high throughput and can successfully identify both known and novel transcripts. However, without precursor targeted enrichment, NGS-based systems may struggle with accurately detecting low-level variants and have reported detection limits between 2–15% variant allele frequency (VAF). Using NGS, Vij et al. found the VAF of *IDH1*-R132H mutations to be 0.8%, a value significantly lower than other clinically relevant mutations in glioma such a *ATRX* [139]. While highly sensitive methods like real-time PCR (as low as 0.0002%) [140] are appealing alternatives for low abundance targets, they typically lack SNV discrimination and lack the high throughput of sequencing-based applications. On the contrary, sequencing-based applications can provide unbiased results for new or previously known SNVs. Additionally, targeted sequencing with user-defined primer sets to enrich specific genes or regions allows for the detection of VAF as low as 0.1–0.2% to be achieved [141]. When using sequencing for diagnostic applications in cancer, whole genome sequencing can be used; however, it often poses a significant time and cost burden. Thus, panels targeting known biomarkers are often employed depending on the cancer type. Panels are beneficial for saving time and costs but also allow for increased sequencing depth [142] and thus, more reliable results. For gliomas, these panels generally target key biomarkers such as *IDH1*, *TP53*, telomerase reverse transcriptase (*TERT)*, and *ATRX* [143,144,145,146,147]. While methods using NGS techniques such as ion-torrent [144,146,147,148] and Illumina [145,149,150] have been developed to detect *IDH* mutations, third-generation techniques such as oxford nanopore technology (ONT) [151,152,153] have recently gained traction as well. These ONT platforms have demonstrated variant detection limits within the general range of those seen in NGS-based systems (3.3%). 

DNA sequencing is commonly used for diagnostic purposes; however, RNA sequencing (RNAseq) assays have also been developed to include *IDH* mutations [154]. RNAseq offers some advantages over DNA sequencing, such as the direct detection of rare splice variants, accurate measurements of gene expression, and detection of non-coding RNA species [155]. RNAseq also serves as a valuable approach toward transcriptome profiling and can provide single-cell resolution [156] and spatial information [157]. RNAseq has the potential to uncover cell-type specific treatment responses, better understand disease mechanisms, and provide quantitative transcriptome-wide data from tissue sections [157]. Using single-cell RNAseq (scRNAseq) of GBM samples, Couturier et al. generated a hierarchal map to uncover therapeutic targets of progenitor cancer stem cells [156]. Specific to *IDH* mutant glioma, scRNAseq has been utilized to explore the identity of progenitor cells for astrocytoma vs oligodendroglioma, as well as identify differences in their respective tumor microenvironments. For example, the data of Venteicher et al. suggest a common progenitor cell type for astrocytoma and oligodendroglioma IDH mutant glioma [158]. While high cost will likely hinder the widespread implementation of RNAseq in a clinical setting in the immediate future, basic research with these techniques may yield valuable targets for novel therapeutics to improve overall survival.

Sequencing is commonly employed with DNA purified from fresh frozen tissues or formalin-fixed paraffin-embedded tissues (FFPE) [143,144,145,146,147,148,149]. Additionally, sequencing for the detection of *IDH* mutations has been successfully performed on cell-free circulating DNA (cfDNA) derived from CSF [159]. Genetic characterization via liquid biopsies has been particularly challenging in gliomas, with low copy number attributed to the blood–brain barrier. Current data supports that the quantity of cfDNA increases with proximity to the tumor [160], with CSF yielding the highest quantity and fluids like blood [161] and urine proving to be more challenging sample types. To this end, amplification-based library preparation methods such as multiplex PCR can be helpful in enriching samples of interest prior to sequencing. Additionally, others have applied Clustered Regularly Interspaced Short Palindromic Repeat (CRISPR) based systems toward the targeted enrichment of regions of interest. An example of this is nanopore Cas9 Targeted Sequencing (nCATS), which selectively ligates sequencing adapters to CRISPR-Cas cut sites by dephosphorylating DNA ends before cleavage occurs, then preferentially ligating to the freshly cut/phosphorylated DNA ends at the cleavage site. nCATS has been utilized to simultaneously determine the *IDH* mutational status and Methyl Guanine Methyl Transferase (MGMT) methylation status in fresh glioma biopsies [162]. CRISPR-based detection of *IDH* mutations has also been utilized directly with CRISPR-Cas12a [163,164]. CRISPR-Cas12a binds DS DNA with high specificity, which then induces non-specific (collateral) cleavage of single-stranded DNA. Various groups have exploited the unique properties of Cas12a by including single-stranded DNA probes that emit fluorescence when cleaved [165,166]. Despite the broad applications of CRISPR-based diagnostics, evidence exists to indicate prevalent nonspecific cleavage of DS DNA by Cas12a [167], and some groups have even attempted to generate variants with more stringent recognition [168].

### 10.2. Epigenetic Detection

As the epigenetic effects of *IDH* mutations ultimately lead to G-CIMP [29], several groups have identified methylome profiling as an alternative method of diagnosis [169,170,171,172,173]. In an analysis of mixed tumor samples, a pairwise similarity heatmap yielded two major clusters—gliomas with and without *IDH* mutations [174]—demonstrating that the characteristic nature of global methylation patterns can be used to differentiate between the *IDH* wildtype and *IDH* mutant genome. Epigenetics can also serve as an indication of changes in the disease state, as the overall methylation level has been found to decrease during progression [171]. Another benefit to methylation profiling is that it can inclusively recognize oncogenic variants of *IDH1/2* and can be further subclassed into tumor types based on epigenetic marks (for example, astrocytoma or oligodendroglioma) [170,175]. Methylation status has been interrogated using Nanopore sequencing [173,176] and methylation profiling arrays [170,171,172,173] and can be performed using snap frozen or FFPE tissue. Nanopore sequencing for the determination of methylation status is particularly advantageous because it does not require a precursor bisulfite treatment, which is associated with DNA degradation [177] and is rapid enough to adapt for intraoperative use [176]. As discussed later, a more aggressive surgical resection for glioma patients with an *IDH* mutation is associated with greater survival benefit [178,179,180], making simple assays capable of performing intraoperatively valuable tools for clinicians seeking to use this information to guide the extent of resection.

### 10.3. Amplification-Based Detection

Amplification methods for the detection of *IDH* mutations are frequently adapted to discriminate between SNVs while retaining the sensitivity necessary for detecting relatively low copy numbers in a high background of non-target nucleic acids. PCR-based amplification is highly characterized, broadly available, and has long been considered a gold standard in clinical practice. The PCR-based detection of *IDH* mutations has been accomplished with several variations and/or adaptations, including digital droplet PCR (ddPCR) [181] qRT-PCR [182], and multiplex PCR coupled to a SNaPshot assay [183]. Another modification of PCR, Beads, Emulsion, Amplification, Magnetics (BEAMing) RT-PCR has also been adapted for the detection of *IDH* mutations [184]. BEAMing PCR is an adaptation of emulsion PCR that meets the need for the detection of low-frequency alleles in a high background of non-target DNA with a sensitivity of as little as 0.01%. This assay has shown promise in detecting *IDH1* mutant mRNA in CSF-derived extracellular vesicles. Through the development of BEAMing-PCR, it was found that the copy number of *IDH1* mRNA was significantly elevated in patients with an *IDH* mutation as opposed to *IDH* wildtype control samples [184], suggesting that a greater focus on mRNA-based detection could be beneficial. 

Relatedly, ddPCR has demonstrated exceptional sensitivity compared to other PCR variations such as qRT-PCR [182,185], and has been employed for the detection of cfDNA in CSF [186] and blood [187,188]. ddPCR is particularly advantageous for the detection of *IDH* mutations due to the water–oil emulsion partitions that allow for high sensitivity in a high background of non-target DNA. A TaqMan-based allele-specific qPCR has also been employed to detect *IDH* mutations in FFPEs and blood [189], further contributing to the possibility of liquid biopsies for patients to infer the *IDH* mutational status prior to surgical resection. Due to the similar survival benefit imparted by *IDH* driver mutations, inclusivity for all known variants with PCR-based assays would be ideal but has proven difficult. Consequently, *IDH1*-R132H has historically been the primary focus as it is the most encountered mutation. Recently, a cartridge-based RT-PCR assay kit (Idylla) has been developed which qualitatively detects five of the most common codon changes for *IDH1* (R132H/C/G/S/L) and nine codon changes in *IDH2* (R140Q/L/G/W and R172K/M/G/S/W). The Idylla assay functions with an input of FFPE has 97% agreement with the sequencing results and requires limited hands-on time for laboratory technicians [190]. With the success of Vorasidenib on less common variants such as *IDH1*-R132C, it has become increasingly urgent for assays to be inclusive and easily implemented. 

In addition to PCR, isothermal amplification techniques have also been developed for the identification and characterization of *IDH* mutations. Loop-Mediated Isothermal Amplification (LAMP) is among the most popular isothermal amplification techniques due to its ability to amplify DNA or RNA in crude cell lysates, eliminating the need for a nucleic acid extraction step. Additionally, LAMP can facilitate the simple and visual interpretation of data through turbidity or colorimetric means. LAMP coupled with a peptide nucleic acid (PNA) clamping probe has been shown to facilitate the detection of *IDH1*-R132H in tumor samples within approximately 1 h [191]. Additionally, a LAMP-based genotyping panel for *IDH1* that uses primers modified with Locked Nucleic Acids (LNA) to mediate SNV specificity has been shown to successfully identify the specific *IDH1* variant present (R132H/L/C/G/S) within 35 min and can be easily interpreted through absorbance or visual interpretation of colorimetric changes [192]. LAMP is particularly advantageous for the detection of *IDH* mutations because of its ability to rapidly amplify DNA in crude cell lysates, making it a viable candidate for intraoperative use.

### 10.4. Histological Detection

IHC using Hematoxylin and Eosin (H&E) stained tissue sections treated with monoclonal antibodies targeting *IDH1*-R132H is commonly employed for the diagnosis of *IDH1* mutations in clinical practice. Samples are typically FFPE or frozen sections; however, frozen sections have been found to yield false positives in some cases [193]. While H09 is the most widely used antibody, others exist [194,195,196], including a more recently developed antibody MRQ-67 which promises comparable sensitivity and specificity but less background [197,198]. Although *IDH1* mutations in glioma are most commonly the *IDH1*-R132H variant, alternative *IDH1*-R132 SNVs, including R132S, R132L, R132G, and R132C have been observed in the molecular analysis of primary tumor samples [20,199] (Table 1). While available monoclonal antibodies specifically target IDH1-R132H, cross reactivity has been observed for other *IDH* SNVs [200]. For example, MsMab-1 recognizes IDH1-R132H/S/G and IDH2-R132S/G, while MsMab-2 recognizes IDH1-R132L and IDH2-R172M [196]. This cross reactivity is not necessarily detrimental, as oncogenic variants of *IDH1/2* mutant are often treated in a similar manner; however, an antibody that is cross-reactive with all oncogenic IDH1/2 variants without exhibiting a positive result for the wildtype would be ideal. IHC is a particularly convenient method of diagnosis for *IDH* mutations from a clinical perspective as the antibody can be applied and assessed alongside routine histological analysis often utilized to gain information about nuclear atypia, mitotic activity, microvascular proliferation, and necrosis within the tumor [201]. More recently, artificial intelligence-based applications such as machine learning and deep learning have advanced the accuracy of histological analysis. To this end, these applications have been highly successful in differentiating between *IDH* wildtype and *IDH* mutant gliomas using digitalized whole slide images [202,203,204,205,206].

### 10.5. D-2-HG as a Surrogate Marker

D-2-HG is generated by hydroxyacid-oxoacid transhydrogenase and/or 3-phosphoglycerate dehydrogenase through side reactions with low catalytic efficiency [208] or as part of an anti-inflammatory response [209]. D-2-HG is subsequently broken down by D-2-HG dehydrogenase [210], maintaining relatively low levels in healthy individuals. Elevated levels of D-2-HG in body fluids can be attributed to the metabolic disorder D-2-Hydroxyglutaric aciduria [211] or oncogenic mutations to *IDH1/2*. To this end, elevated levels of D-2-HG may be utilized as a surrogate marker as an alternative to the genetic or histological elucidation of *IDH* mutational status. 

D-2-HG can accumulate in *IDH* mutant tumors at a median concentration of 1,965.8 µM [117] and at elevated levels in patient CSF [212,213,214] (up to 109.0 µM), and blood [212,215,216,217] (up to 10.9 µM). Data obtained from currently available studies show a clear and positive correlation between D-2-HG elevation in patients with *IDH* mutants in comparison to their *IDH* wildtype counterparts; however, the ideal sample type has been a topic of some debate. The ability to quantify D-2-HG in body fluids as a surrogate marker of *IDH* mutational status could revolutionize the current standard of care and allow for preoperative, intraoperative, and/or postoperative characterization. Harnessing D-2-HG for diagnostics could also facilitate remote testing for patients, as it is highly stable following exposure to excess heat [218] or several freeze–thaw cycles [213]. A current limitation of this approach is the availability of a simple, rapid, and user-friendly method for quantification. At this time, D-2-HG can be detected by liquid chromatography–mass spectroscopy (LC–MS) [212,214], gas chromatography–mass spectroscopy (GC–MS) [219] or magnetic resonance spectroscopy (MRS) [220]. Fluorometric and colorimetric kits are also commercially available for this purpose [221], but hands-on time and price are likely limiting factors for clinical implementation. A fluorescent resonance energy transfer (FRET)-based biosensor for the detection of D-2-HG has been developed [222] but is limited by its dynamic range and use at a physiologically impossible pH of 10. A more recent FRET-based biosensor has been validated using glioma tumor samples and contrived clinical specimens, demonstrating quantification of D-2-HG within a clinically relevant range (~300 nM–100 µM) at physiological pH [223]. FRET-based sensors are promising diagnostic tools; however, they have not yet been evaluated in a clinical setting.

While highly useful for diagnostic purposes, the use of D-2-HG as a surrogate marker can also provide valuable insight into the effectiveness of pharmaceutical mutant IDH inhibitors such as Ivosidenib [224,225]. This utility becomes an even greater point of interest with the outstanding efficacy of Vorasidenib [10], which is expected to receive a fast track for approval by the Food and Drug Administration (FDA). A convenient, accurate, and highly characterized tool for the quantification of D-2-HG could also provide novel insights into the relationship between D-2-HG with *IDH* mutant cancers such as the correlation between disease burden and D-2-HG level. Additionally, the use of D-2-HG as a surrogate marker would offer significant benefits for earlier implementation of treatment as it could alert clinicians of an *IDH* mutation without the need for a tissue biopsy. Another benefit to a liquid biopsy approach is that it would be expected to indirectly identify the presence of all oncogenic *IDH1/2* mutations due to the conserved accumulation of D-2-HG across variants [226]; however, data in this respect is scarce due to the rarity of non-*IDH1*-R132H variants in glioma and the relatively small sample sizes of currently available studies. Future studies to validate the consistency of D-2-HG as a surrogate marker of IDH variants would be highly desirable in determining its value as a comprehensive indicator of disease. Moreover, surveying the levels of D-2-HG in a large cohort of IDH mutant glioma patients in various body fluids preoperatively, postoperatively, and through the process of treatment is needed to determine correlations, if any, with D-2-HG levels and disease burden. 

In addition to liquid biopsy approaches, MRS has also shown great promise for the non-invasive detection of 2-HG, which we discuss below.

### 10.6. MRI

Conventional MRI is an important standard of care for gliomas, and radiologic features such as contrast enhancement and multifocality can be useful in determining the molecular characteristics of a tumor [227]. Contrast enhancement is utilized to visualize the vascularity of a structure through the administration of a contrast agent, while multifocality is the presence of multiple distinct lesions. Imaging characteristics including a higher percentage of non-contrast enhancing tumors, larger tumor sizes, the presence of cysts, and the presence of satellites have been shown to predict the presence of an *IDH* mutation with 97.5% accuracy [228]. Furthermore, *IDH* mutant tumors typically grow in a single lobe, with the most common being the frontal [229] or temporal lobe, whereas *IDH* wildtype tumors are frequently distributed between lobes [230]. Overall, the use of MRI for the elucidation of *IDH* mutational status is beneficial due to its non-invasive nature, allowing for earlier diagnosis than methods such as sequencing or IHC which require a tissue biopsy. T2-weighted imaging is one of the most common contrast sequences in MRI, and T2-weighted Fluid-attenuated inversion recovery (T2-FLAIR) is an advantageous approach because it can be performed using only standard MRI sequences to differentiate between *IDH* wildtype and *IDH* mutant astrocytomas. T2-FLAIR MRI sequences enhance the contrast between gray matter and white matter to improve the visibility of lesions [231], and the use of T2-FLAIR mismatch for detecting *IDH* mutant gliomas is based on T2 complete (or mostly complete) homogeneous hyperintense signal and attenuation of FLAIR signal intensity with a bright peripheral rim [232,233]. Additionally, the non-contrast enhancing properties of *IDH* mutant gliomas make the T2-FLAIR mismatch sign a useful indication of *IDH* mutational status [234].

Perfusion-weighted MRI (PWI) has recently gained attention for facilitating a more accurate determination of tumor grade in comparison to conventional MRI alone [235]. PWI provides information about tissue vascularization and angiogenesis [236], and the relative cerebral blood volume (rCBV). rCBV is a value that can be calculated from PWIs by determining the volume of blood in a specific quantity of brain tissue. rCBV values can serve as a powerful indication of *IDH* wildtype vs *IDH* mutant gliomas, as recent studies have found these values to be 2–2.5 higher in *IDH* wildtype glioma samples than their *IDH* mutant counterparts [237]. PWI has also been combined with dynamic susceptibility contrast-enhanced MRI (DSC) [229]. DSC utilizes signal loss induced by paramagnetic contrast agents like gadolinium-based compounds on T2-weighted images to determine additional parameters such as relative cerebral blood flow (rCBF) and mean transit time (MTT). These parameters are subsequently used to assess regional perfusion and have been used in combination with PWI to determine the apparent diffusion coefficient (ADC). To this end, the minimum/relative ADC [229] and mean ADC [238] have been found to be significantly elevated in *IDH* mutant astrocytoma (grade II and III) compared to *IDH* wildtype tumors. These findings are supported by the use of diffusion tensor imaging (DTI) to determine the ADC of gliomas, which found that fractional anisotropy and ADC from DTI can successfully determine the *IDH1* mutational status in gliomas [239]. Diffusion-weighted MRI (DWI) techniques allow for the observation of the cellular architecture of tumors and surrounding tissue [240] by assessing the Brownian motion of water molecules and have shown promise in differentiating between *IDH* wildtype and *IDH* mutant gliomas [239,241]. DWI and PWI have also been used to assess the response of *IDH1* mutant gliomas to pharmaceutical *IDH* mutant inhibitors, where an increase in the normalized rCBV and ADC were found to be a useful indicator of antitumor response within a timeframe of 2–4 months [242]. Recently, artificial intelligence applications such as machine learning and deep learning have become valuable tools for highly accurate differentiation between *IDH* wildtype and *IDH* mutant gliomas in MRI-based applications [243,244,245,246,247,248]. Carosi et al. have also extensively covered MRI-based techniques for the detection of *IDH* mutant gliomas and other solid tumors [249].

### 10.7. MRS

Due to the unique accumulation of D-2-HG, *IDH* mutations may also be detected with the use of MRS through the measurement of total 2-HG [250,251,252,253,254,255]. With an in vivo sensitivity of approximately 1 mM and an ex vivo sensitivity for intact biopsies of 0.1–0.01 mM [256], MRS offers localized 2-HG quantification directly in the lesion and is well poised to differentiate between *IDH* wildtype and *IDH* mutant gliomas. D-2-HG is known to accumulate in *IDH* mutant gliomas at a median concentration of 1965.8 µM, a value heavily contrasted by IDH wildtype gliomas which yield a median value of 14.0 µM [117]. 2-HG MRS has demonstrated impressive sensitivity and specificity that outperform conventional MRI as well as DWI and PWI [251]. Additionally, as D-2-HG is known to deplete in response to treatment with pharmaceutical inhibitors such as Ivosidenib [224,225] or Vorasidenib [10], MRS is currently being explored for its ability to gauge the effectiveness of mutant IDH inhibitors [257]. While MRS is both highly sensitive and specific, this method may be limited by a low signal-to-noise ratio which requires lesions to be at least several milliliters in volume and sufficiently distant from fluid–brain or air–fluid interfaces. Furthermore, MRS cannot discriminate between D-2-HG and its naturally occurring enantiomer, L-2-HG. L-2-HG is not a useful indication of disease state; thus, enantiomer discrimination would add a greater level of confidence to MRS-based applications. However, analysis of total 2-HG within glioma tissues has shown a median value of 1971.5 µM for IDH mutant, compared to a median value of 27.0 µM for IDH wildtype [117]. These results still provide a clear indication of the disease state; however, including larger patient populations in these types of studies is necessary to determine how much variability exists in enantiomer ratios and 2-HG concentrations.

## 11. Clinical Implications of *IDH* Mutations 

### 11.1. Clinical Classification of Gliomas

Despite advances in diagnosis and treatment of glioma, the prognosis for the disease is poor [258]. To aid in curbing the lack of improvement in life expectancy, the medical community has increased reliance on molecular analysis of tumors. In addition to the traditional histological and immunohistological methodologies used to characterize brain tumors, the 2021 WHO CNS5 integrated molecular diagnostics for the further classification of tumors [3]. As a result, there are six new families in the classification of gliomas including the following: (1) Adult-type diffuse gliomas, (2) Pediatric-type diffuse low-grade gliomas, (3) Pediatric-type diffuse high-grade gliomas, (4) Circumscribed astrocytic gliomas, (5) Glioneuronal and neuronal, and (6) Ependymomas. The most common family of tumors, adult-type diffuse gliomas, includes GBM and mutant *IDH* gliomas [259]. With this molecular classification, *IDH* mutant gliomas have been identified as a biologically distinct group of tumors. In cases of adult-type diffuse glioma, the most significant molecular prognostic indicator is *IDH* mutational status, where *IDH* mutant gliomas are less aggressive than their wildtype counterparts [260]. Molecular diagnosis is imperative as *IDH1/2* mutations are associated with a prolonged survival benefit of approximately 4-fold when molecular identification is combined with surgical resection [8,261]. Identification of *IDH* status stratifies adult-type diffuse gliomas into separate classifications where GBM is exclusively characterized as *IDH* wildtype. Further molecular classification for wildtype GBM includes identification of one or more of the following: mutations found in the *TERT* promoter, amplification of epidermal growth factor receptor (*EGFR*) and chromosome 7 gain (partial or complete)/chromosome 10 loss [262,263,264,265].

Separate from *IDH* wildtype GBM, mutant *IDH* gliomas are molecularly categorized into oligodendroglioma or astrocytoma. The hallmark of oligodendroglioma includes the 1p/19q codeletion, whereas *ATRX* and *p53* mutations differentiate mutant *IDH* gliomas into astrocytomas. Among gliomas with *ATRX* loss, 89% retained *IDH1/2* mutations while *ATRX* retention in *IDH1/2* mutants was strongly associated with 1p/19q loss, making this a differentiating feature associated with oligodendroglioma [266]. Further molecular characterization of the mutant *IDH* astrocytoma includes detection of the presence or absence of the *CDKN2A/B* gene. Homozygous deletion of the *CDKN2A/B* gene characterizes the tumor as grade 4. Grading of mutant *IDH* astrocytomas retaining the *CDKN2A/B* gene relies on histological analysis to be differentiated into grade 2 and 3 mutant *IDH* astrocytomas. It is worth noting that astrocytomas can also be categorized as wildtype *IDH* but also retain one or more of the *TERT* promoter mutations, *EGFR* amplification and/or chromosome 7/10 aneuploidy. The classification of gliomas has also been extensively covered in a recent review by Weller et al. [267].

### 11.2. Influence of Mutational Status on the Production of D-2-HG

Mutant *IDH* astrocytoma has an incidence rate of 0.44 per 100,000 individuals with approximately 3000 cases identified in the United States, making up 11% of all diffuse gliomas [259]. From a clinical perspective, *IDH* mutant gliomas have a significant survival advantage over wildtype gliomas. As identified earlier in this review, *IDH1/2* mutations lead to the accumulation of D-2-HG which has pleiotropic oncogenic effects that result in prolonged life expectancy and delayed therapeutic interventions. The overwhelming majority of this data is based on the *IDH1*-R132H mutation. However, very little is known regarding the cellular accumulation of D-2-HG for the less common mutations of *IDH1* such as R132C/G/S/L. A recent publication by Pusch et al. performed a study evaluating the enzymatic activity of recombinant IDH1 variants on isocitrate/αKG substrate and found that the prevalence rate of *IDH1*-R132X variants (Table 1) found in patients is inversely proportional to their respective affinities suggesting that selective pressure (i.e., D-2-HG toxicity) favor the common *IDH1*-R132H variant [226]. Based on these data, a favorable clinical outcome would be seen due to increased production of the oncometabolite, D-2-HG. Natsumeda et al. demonstrated that elevated 2-HG had a better overall survival than lower 2-HG [268]. However, this analysis was performed using MRS in the evaluation of the total 2-HG. As seen in multiple studies, *IDH1* mutant clinical patient outcomes are variable [260,269,270,271,272]. Of significance to the prognostic implications of *IDH* mutants is the presence of sub-clonal populations with mosaic expression [273,274,275,276,277,278,279,280], confounding interpretation of the role of D-2-HG production. As a result, quantitative analysis of sub-clonal populations may provide more clarity for the treatment of *IDH1* mutant gliomas. 

### 11.3. Pharmaceutical Treatment of IDH Mutant Gliomas

Mutant *IDH* gliomas and subsequent D-2-HG production provide a highly druggable target due to its role in glioma formation and progression [281]. Early *IDH* inhibition is especially important, as tolerable drugs could potentially delay the long-term neurocognitive toxicities of standard treatment which has been identified as a detrimental factor in employment and quality of life [282,283]. Mutant IDH inhibitors have been investigated in glioma for approximately ten years [284,285]. Two inhibitors, Ivosidenib and Enasidenib, have been approved by the FDA for the treatment of *IDH*-mutant leukemia. The two most studied drugs in glioma are Ivosidenib (AG-120), a mutant IDH1 inhibitor, and Vorasidenib (AG-881) an IDH1/2 inhibitor [10]. Ivosidenib is a specific, reversible, allosteric competitive inhibitor of mutant IDH1, and has shown clinical utility in treating *IDH1*-mutant gliomas [10]. Vorasidenib, a pan-IDH1/IDH2 inhibitor, displays CNS penetration and successfully demonstrated itself as a potential treatment for *IDH*-mutant glioma pending FDA approval [10]. The recent phase 3 INDIGO trial evaluating Vorasidenib demonstrated significantly prolonged disease stability as well as the ability to delay additional standard therapeutic interventions [10]. More recently, a dual-inhibitor of both nicotinamide phosphoribosyl transferase (NAMPT) and mutant IDH1 has been developed which has the ability to cross the BBB and demonstrates potent efficacy in vivo [286]. Carosi et al. have recently provided an in-depth analysis of pharmaceuticals targeting IDH-mutant gliomas and other solid tumors [249].

In addition to pharmaceuticals that directly inhibit the mutant *IDH* protein, inhibitors that interact with proteins associated with epigenetic marks also make excellent targets. Specifically focusing on the acetylation and methylation state of chromatin, HDACs and Jumonji class demethylases are chromatin erasers that are of clinical interest. In *IDH* mutant glioma, the downstream effect of D-2-HG production inhibits αKG-dependent DNA demethylases rendering chromatin hypermethylated. HDACs are enriched in hypermethylated regions of chromatin, potentially making mutant *IDH* cells more susceptible to HDAC therapy. The HDAC inhibitor Panobinostat (Farydak) was an FDA-approved drug for the treatment of multiple myeloma that had been recognized as a potential *IDH* mutant inhibitor for gliomas [287]. Following treatment with Panobinostat, *IDH1* mutant glioma cells demonstrated increased cytotoxicity and inhibited proliferation [45]. Panobinostat was also found to compound to inhibit growth in *IDH1* mutant glioma lines [46]. Panibostat was recently withdrawn from USA markets due to incomplete post-approval studies and the inability to confirm the clinical benefits within the given constraints [288]. Of significance to Jumonji class demethylases, KDM5 has been shown to be a target of 2-HG production resulting in the inhibition of lysine demethylase activity and contributing to cellular transformation in mutant *IDH* glioma [36]. Nie et al. have described a family of pyrazolyl pyridines that have demonstrated potent activity targeting KDM5A/B resulting in an increase in H3K4me3 epigenetic marks in cancer cell lines [289]. 

### 11.4. Molecular Basis for the Improved Prognosis of IDH Mutant Gliomas

The survival advantage for gliomas retaining *IDH* mutations is poorly understood; however, potential mechanisms for this benefit are slowly being elucidated. *IDH* mutant cells demonstrate a greater degree of hypermethylation in undifferentiated neural progenitor cells than in mature astrocytes, suggesting that the epigenetic modifications within the IDH mutant genome are dependent upon cellular context [90]. Pursuant to this point, a recent publication identified that D-2-HG reduces glioma cell growth by inhibiting the m^6^A epi transcriptome regulator, FTO [81]. The aforementioned enzyme is responsible for m^6^A hypermethylation for a specific set of mRNA transcripts including *ATF5* (Activating Transcription Factor 5) leading to increased cellular apoptosis. Interestingly, inhibition of FTO led to the growth characteristics of wildtype *IDH* gliomas to be more consistent with *IDH* mutant growth phenotype. Within *IDH* mutant astrocytomas, global DNA methylation status and CDKN2A homozygous deletion were found to be significant prognostic indicators [290]. Non-canonical mutations also play a role as a prognostic tool for gliomas. Interestingly, non-canonical *IDH1*-R132 mutations have an improved prognostic outcome compared to the canonical *IDH1*-R132H mutation in gliomas [291]. This lends support for the development of diagnostic tools capable of readily differentiating the mutational status in patients. 

### 11.5. Clinical Trials

A comprehensive assessment of current trials targeting mutant *IDH* gliomas can be found in reviews by Sharma and Kayabolen [292,293].

## 12. Conclusions

*IDH* is the most common metabolic mutation associated with oncogenesis, and the production of D-2-HG yields a unique cancer phenotype that includes a characteristic epigenetic profile. *IDH* serves as a critical biomarker in hematological malignancies as well as solid tumors such as glioma, chondrosarcoma, and cholangiosarcoma. While we focus solely on gliomas in this review, Carosi et al. recently covered the role of *IDH* mutations shared by or unique to these solid tumors. The epigenetic, immunological, and metabolic characteristics are highlighted, as well as selected diagnostics and pharmaceuticals [249]. In this review, we expand upon this work to provide a highly detailed analysis of the molecular mechanism and consequences of *IDH* mutant gliomas with a special focus on the role of metabolism. D-2-HG is essential for the development and maintenance of *IDH* mutant glioma. Its inhibition of *α*KG-dependent dioxygenases yields a unique epigenetic profile/transcriptome reliant upon its continued production and accumulation. We provide an extensive review of current data highlighting the specific role of D-2-HG in various pathways, including m^6^A modification, DNA damage repair, metabolic rewiring, and epigenetics. In addition to the molecular mechanism of *IDH* mutations in glioma, we also extensively cover the past, present, and future directions of diagnostics that have been developed for the detection of *IDH* mutant gliomas. With the recent success of Vorasidenib [10], innovative methods capable of early detection will be increasingly imperative for improving patient prognosis. 

## Figures and Tables

**Figure 1 biology-13-00885-f001:**
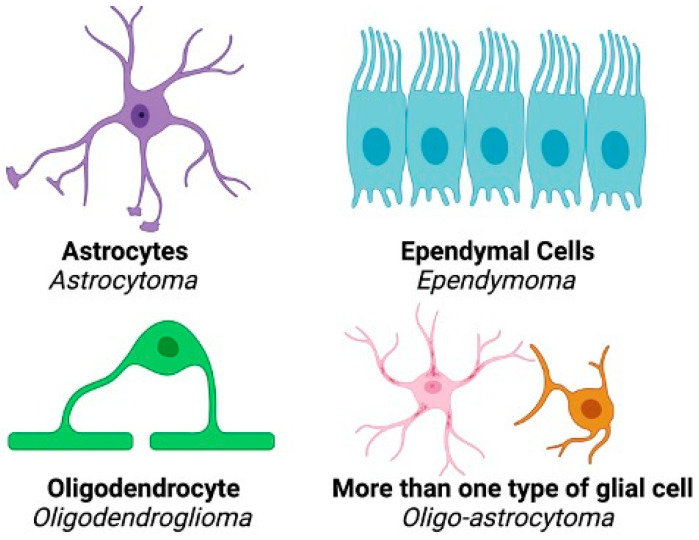
Examples of neuroglial cells and glioma types resulting from malignant transformation. Glial cells support neuronal processes and maintain the surrounding environment. This figure was created with biorender.com.

**Figure 2 biology-13-00885-f002:**
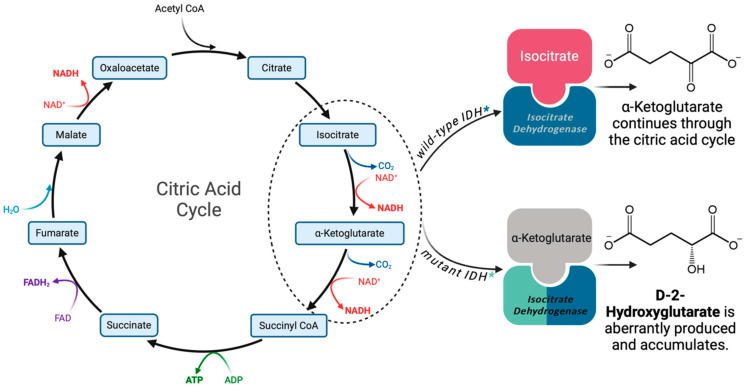
The difference in roles between wildtype and mutant IDH within the citric acid cycle. Wildtype IDH catalyzes the conversion of isocitrate to *α*KG. Mutations to conserved residues within the catalytic site of IDH elicit a gain-of-function that results in the aberrant conversion of *α*KG to D-2-HG. This figure was created with biorender.com.

**Figure 3 biology-13-00885-f003:**
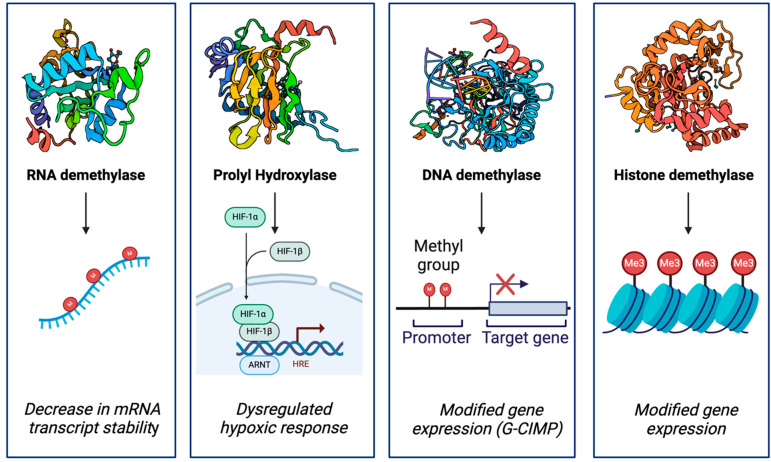
D-2-HG competitively inhibits various *α*KG-dependent dioxygenases. From (**left**) to (**right**): YTHDC1 with N^6^-Methyladenosine (m^6^A), PHD2, TET2 in complex with DNA, the Jumonji domain of human Jumonji domain containing 1C protein. Crystal structures were sourced from Protein Data Bank (PDB); this figure was created using Biorender.com.

**Figure 4 biology-13-00885-f004:**
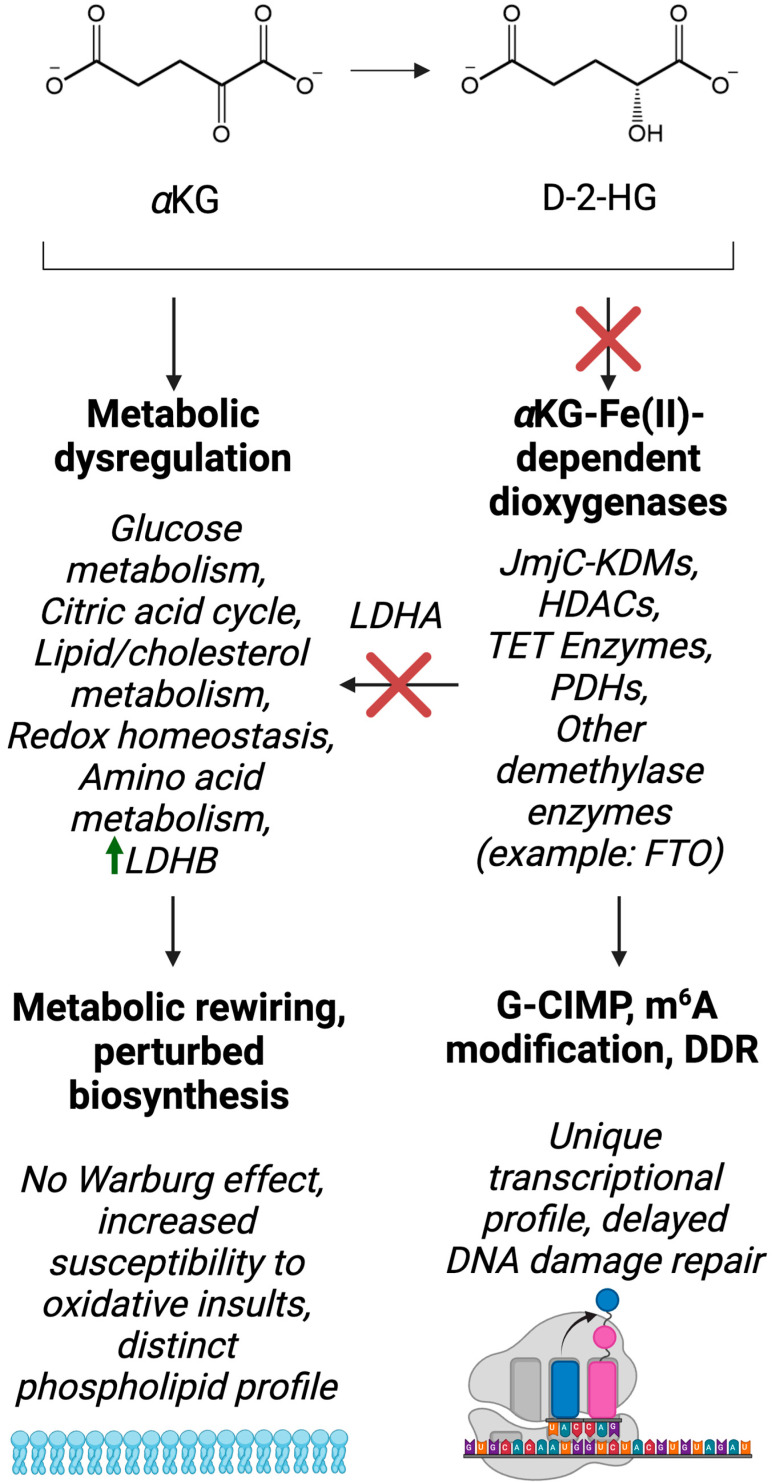
The molecular impacts of *IDH* mutations discussed in this review. Metabolic rewiring and D-2-HG mediated inhibition of *α*KG-Fe(II)-dependent dioxygenase play important roles in the unique phenotype of *IDH* mutant cells. This figure was created with Biorender.

**Table 1 biology-13-00885-t001:** Frequency of *IDH* variants in low-grade glioma based on values reported in the current literature [199,207].

*IDH* Variant	Frequency in Low-Grade Glioma (%)
*IDH1*-R132H	62.0–93.0
IDH1-R132C	2.9–4.3
*IDH1*-R132G	1.0–2.5
*IDH1*-R132S	1.1–2.2
*IDH1*-R132L	0.2–0.6
*IDH2-*R172K	2.8–3.0
*IDH2-*R172W	0.6
*IDH2-*R172M	0.8
*IDH2-*R172S	0.2
*IDH1-*R132P *IDH1-*R132V *IDH2-*R172T	Extremely rare

## Data Availability

The original contributions presented in the study are included in the article. Further inquiries can be directed to the corresponding author.

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
