# Peer review of "IDH Mutations in Glioma: Molecular, Cellular, Diagnostic, and Clinical Implications"

_biology, 2024, doi:10.3390/biology13110885_

Round 1

Reviewer 1 Report

Comments and Suggestions for Authors

IDH mutations in glioma: molecular, cellular, diagnostic, and clinical implications.

The manuscript is a well-researched and provide an in-depth detailed review of IDH mutations in gliomas. However, it can be improved by refining sentence structure for clarity and expanding on specific mechanisms and diagnostic techniques. Including visual diagrams, tables, will enhance the paper's readability and provide a more comprehensive understanding.

Overall

·         There are many long sentences, consider breaking them into shorter, more concise statements. This will improve readability and ensure that key points are clearly communicated.

·         The diagnostic section lacks an evaluation of the sensitivity, specificity, and limitations of each method discussed.

·         Please discuss some mechanisms by which IDH mutations confer a survival advantage in gliomas.

·         Including a diagram of the molecular pathways affected by IDH mutations would enhance the paper. These could help readers grasp complex interactions more quickly.

·         Ensure that all abbreviations are defined and listed in alphabetical order.

·         Tables wherever necessary would help to clarify outcomes and statistical data more effectively. Tables summarizing key data, such as the prevalence of IDH mutations or the performance of diagnostic techniques, would be useful.

Specific

·         Line 54: The shift from a general discussion on gliomas to IDH mutations feels abrupt. It could be smoothed out by linking the general characteristics of gliomas directly to the relevance of IDH mutations in this context. Ensure that all cited references are relevant when discussing the discovery of IDH mutations and their role in gliomas.

·         Line 77: missing reference, key information.

·         line 110: genome-wide

·         line 123-124: how do these modifications correlate with transcriptional outcomes in cancer?

·         line 138: which we address later in this review later**

·         line 143: remove actively

·         line 147-148: including three of the six HDAC genes that are expressed in gliomas

·         The connection between HDAC and IDH mutant cancers is mentioned but could be elaborated for clarity. Specifically, how HDAC influence hypermethylation, and what are the implications for gene expression in IDH mutant gliomas?

·         To include has been used a lot of time, line 154 and line 159, consider rephrasing

·         line 155-156: please mention the biochemical function of TET enzymes? and what is the significance of these oxidized derivatives in cancer.

·         line 159-160: Missing reference “TET enzymes are often mutated in various types of cancer to include secondary AML…”

·         line 192: The phrase "or in some cases inflammatory conditions" could be changed to "including under certain inflammatory conditions" for better clarity.

·         line 230: how IDH inhibitors specifically affect the m6A modification process? in 1 sentence.

·         line 242: what was the relation between ATF5 in the context of IDH mutant glioma?

·         line 255 onwards: seems there is a change in font size maybe?

·         line 259-260: Missing references “Isocitrate is subsequently isomerized into citrate, which is broken down by ATP citrate lyase into acetyl-CoA. Acetyl-CoA can be used for fatty acid biosynthesis under these condition”

·         line 301-302: Citric acid cycle was repeated twice, maybe writing Anaplerotic reactions replenish citric acid cycle intermediates, maintaining cycle flux.

·         line 366-367: The sentence is too big. Consider breaking into 2 separate sentences. Maybe writing, this sensitivity is partly due to the inhibition of αKG-Fe (II)-dependent dioxygenases such as ALKBH2 and ALKBH3.

·         line 370-371: “…and at clinically relevant concentrations and ratios of D-2-HG/αKG”. What are the conc. and ratios?

·         line 393-401: the overall catch of the line is not clear. Discuss the conflicting results concerning IDH mutant cells' response to radiation-induced DNA damage in a clearer and easy way.

·         line 416-418: Consider breaking this long sentence for clarity.

·         line 419-423: please add a brief explanation of how sequencing surpasses traditional PCR in sensitivity/specificity, particularly in detecting low-frequency mutations in heterogeneous tumors.

·         line 432-433: Expanding on the advantages and applications of RNA sequencing for IDH mutations would help the discussion, considering the importance of transcriptomic in understanding tumor biology.

·         line 443-450: Is there a potential off-target effects of CRISPR techniques in clinical diagnostics of IDH mutations.

·         line 561-564: Highlight the research gaps, such as the need for larger studies to validate the consistency of D-2-HG as a marker across all oncogenic IDH mutations and suggest future research directions.

·         line 568-572: Why do IDH mutant tumors typically grow in a single lobe? Could it be linked to their metabolic or growth characteristics. Make sure that all technical jargon and MRI techniques are clearly written, to make the content easy to readers who may not be familiar with it.

·         line 608-610: While detailing the sensitivity levels of MRS, discuss the typical range of D-2-HG concentrations found in IDH mutant versus wildtype gliomas. Highlight the limitations of MRS and discuss ongoing research that might overcome these challenges.

Sentence 624-626: While discussing the poor prognosis of glioma, the author could include specific statistics or recent findings that highlight the improvements, if any, that have been achieved through molecular diagnostics.

Comments on the Quality of English Language

The manuscript is generally well-written, but there are areas where the quality of English could be improved to enhance clarity. Sometimes there are overgeneralization, and sometimes there is overly long sentences.

Author Response

We thank this reviewer for the many thoughtful comments. Please see below for how we have addressed each one:

There are many long sentences, consider breaking them into shorter, more concise statements. This will improve readability and ensure that key points are clearly communicated.

We went through our manuscript and broke up several sentences that were quite long. Thank you for this suggestion. 

The diagnostic section lacks an evaluation of the sensitivity, specificity, and limitations of each method discussed.

There is now a paragraph that focuses on sensitivity/specificity and limitations from 496-513

Please discuss some mechanisms by which IDH mutations confer a survival advantage in gliomas.

We have added a section from 811-825 that focuses on this topic. 

Including a diagram of the molecular pathways affected by IDH mutations would enhance the paper. These could help readers grasp complex interactions more quickly.

This has been included and is now Figure 4.

Ensure that all abbreviations are defined and listed in alphabetical order.

A table has been added (pg 24-25) which includes all abbreviations defined and listed in alphabetical order. 

Tables wherever necessary would help to clarify outcomes and statistical data more effectively. Tables summarizing key data, such as the prevalence of IDH mutations or the performance of diagnostic techniques, would be useful.

A table summarizing the prevalence of IDH mutations is 'Table 1' split between pages 18 and 19.

All minor comments listed under 'specific' have been individually addressed. Thank you again for your time and feedback.

Reviewer 2 Report

Comments and Suggestions for Authors

The recent publication on a similar topic has already been released, diminishing the novelty and originality of the findings presented in this review.

Author Response

The reviewer did not specify which publication they are referring to; thus, we are unable to respond to this comment in a productive manner.

Reviewer 3 Report

Comments and Suggestions for Authors

Excellent review.  It is comprehensive and very well-written.  There is a lot of information, which makes it a challenge to read straight through; however, it is a valuable contribution to the literature as a very complete review of IDH mutations. The references cited are very extensive (273!).  There is no apparent bias in the references used

Minor changes that are suggested prior to approval include that panobinostat has been pulled from the market and is not FDA approved any longer (as of March 24,

Author Response

We thank this reviewer for their kind words and enthusiasm.

We have modified the manuscript (line number 805-806) to specify that Panibostat was withdrawn from USA markets. 

Round 2

Reviewer 1 Report

Comments and Suggestions for Authors

Line 126: STAT (signal transducer and activator of transcription)

Line 213: "Fe(III) to Fe(II), a required cofactor of TET." should be changed to "Fe(III) to Fe(II), an essential cofactor of TET."

Author Response

Thank you for these suggestions. Both are now changed in the manuscript. 

Reviewer 2 Report

Comments and Suggestions for Authors

Thank you so much for giving me this opportunity to review this article. 

In reply of Author Response (The reviewer did not specify which publication they are referring to; thus, we are unable to respond to this comment in a productive manner).

Again I have written the latest review article citation here which is on the same topic and that is the big reason to support my decision.

Carosi F, Broseghini E, Fabbri L, et al. Targeting Isocitrate Dehydrogenase (IDH) in Solid Tumors: Current Evidence and Future Perspectives. Cancers (Basel). 2024;16(15):2752. 

Author Response

Thank you for providing the citation for this review article- we were previously not aware of it. We have now cited this publication several times in our manuscript:

1720-1721: we reference this paper for their extensive coverage of MRI-based detection of IDH mutations in solid tumors 

1905-1906: we reference this paper for its detailed section on pharmaceuticals targeting IDH mutant proteins

1992-1994: we make a point in the conclusion section to point out that Carosi et al recently released a review on IDH mutations, and highlighted some key differences between our review and theirs.

Again, thank you for pointing this out to us so that we may cite this group and give them credit for their work.